# Effects of Broad Bean Diet on the Growth Performance, Muscle Characteristics, Antioxidant Capacity, and Intestinal Health of Nile Tilapia (*Oreochromis niloticus*)

**DOI:** 10.3390/ani13233705

**Published:** 2023-11-29

**Authors:** Xiaogang He, Haoming Shu, Tian Xu, Yuting Huang, Jiajun Mo, Chunxiang Ai

**Affiliations:** 1College of Ocean & Earth Sciences, Xiamen University, Xiamen 361005, China; hexiaogang2023@126.com (X.H.); shuuming@163.com (H.S.); 22320211151349@stu.xmu.edu.cn (Y.H.); 2Anyou Biotechnology Group Co., Ltd., Suzhou 215437, China; mojiajun2022@163.com; 3Marine and Fishery Institute of Xiamen, Xiamen 361008, China; tian.xu@my.jcu.edu.au

**Keywords:** broad bean, crisp, Nile tilapia, intestinal health, muscle

## Abstract

**Simple Summary:**

Two types of diets with broad bean added (additive amounts of 40% and 50%, respectively), and the other two diets further added with the crisping functional package, were tested for the crisping effect on Nile tilapia. They all reached the crisping effect but had no adverse effect on the growth performance, and the supplement of the crisping functional package can further improve the crisping effect (especially the G5 diet). In addition, the crisping functional package also helped to improve the intestinal tissue and microbiota structure.

**Abstract:**

Four crisping diets were designed to conduct a feeding experiment to investigate the use of broad bean in the formulated feed of Nile tilapia and optimize its use. The growth performance, muscle characteristics, antioxidant capacity, and intestinal health of Nile tilapia with an initial body weight of 617.32 ± 1.64 g were evaluated after being fed with different diets for 90 days. The results showed no adverse effect on the growth performance of Nile tilapia fed with broad bean diets. Contrastingly, some improvements were found in WGR and SGR, but a lower FCR was obtained. The supplementation of broad beans weakened the antioxidant capacity of fish but did not influence liver health and the immune system. Increasing the amount of broad bean addition can increase muscle quality values, and an embrittling functional package being added to the diet can also improve muscle hardness, adhesiveness, and chewiness of Nile tilapia muscle. In addition, the crisping functional package can be applied to tilapia crisping formulated feed, which helps to improve the integrity of the intestinal tissue structure and optimize the intestinal microbiota of Nile tilapia. The final achievement of this study is to provide a theoretical reference for optimizing the breeding technology of crispy Nile tilapia and developing a specialized crisping diet for the species.

## 1. Introduction

Tilapia belongs to the family Cichlidae (order Perciformes), and it is the second most important farmed finfish in the world [1]. As an excellent worldwide species of aquaculture designated and recognized by FAO (Food and Agriculture Organization of the United Nations), tilapia is mainly native to Africa and has been introduced into over 100 countries or regions for aquaculture to improve the local protein supply [2,3]. Nile tilapia (*Oreochromis niloticus*) is the most commonly farmed and introduced tilapia species, producing up to 4.6 million tons in 2019 [4]. However, the market value of tilapia stays low, so it is necessary to search for a solution to improve the quality of fish products [5]. One possible solution could be to improve the flesh quality by increasing the textural mechanical properties of flesh, such as hardness and chewiness [6]. Previous studies have shown that dietary components are a significant factor in affecting flesh quality in fish [7].

The first application was taken with another aquaculture species, grass carp (*Ctenopharyngodon idellus*). The process of flesh crisping in grass carp is an emerging aquaculture product named “crispy carp”, which is obtained by replacing the normal fish feed with broad beans (*Vicia faba* Linn) for about 120 days. Compared to ordinary grass carp, this crispy flesh had higher hardness and chewiness; such features were welcomed by more and more consumers and effectively increased the economic value of grass carp [8]. As tilapia are omnivorous fish species with a higher dietary carbohydrate-utilizing ability than other carnivorous fish, they show great potential for digesting plant-sourced ingredients [9]. Hence, broad beans are also expected to perform a crisping handle on Nile tilapia.

However, two major issues impede the application of broad beans in aquaculture. The pretreatment of broad beans is a labor-intensive procedure, which limits the scale and efficiency of production; additionally, these beans contain high antinutritional factors and an incomplete amino acid profile in the beans [10]. Thus, some researchers investigated different crisping options, such as preparing compound fish feeds [11]. On the other hand, researchers can address the shortcomings of the amino acid compositions of broad beans by the development of complementary feed formulas.

The mechanism of fish flesh crisp has also been well-studied on grass carp. It is generally accepted that the process of fish meat becoming crispy is mainly a change in the physical and chemical properties of fish muscle fibers, which is reflected by an increase in muscle fiber density, a decrease in muscle fiber diameter, and an increase in the collagen content of the flesh [12]. Previous research suggested that muscle fiber density is negatively correlated with muscle fiber diameter but positively correlated with both muscle tissue hardness and chewiness. In addition, the collagen content of fish meat is positively correlated with muscle hardness [13,14]. There were several studies about using broad beans as a replacement for other plant proteins in the diet of Nile tilapia, in which growth performance and textural quality of fish were tested [14,15,16]. However, detailed studies on the effects of other aspects of fish, such as immune status and intestinal microbial community structure, are deficient. In fact, related research on crispy tilapia is even much slower than that of grass carp.

Therefore, this study aims to investigate the use of broad beans in the formulated diet of Nile tilapia and further optimize the broad bean diet. Growth performance, serum physiology and biochemistry, histological morphology, muscle characteristics, and intestinal microbiome were evaluated to determine the effects of the broad bean supplement. Moreover, the results of this study provide a theoretical reference for crisping diet development and precise nutritional regulation of crispy Nile tilapia. This information will also promote the healthy and sustainable development of the Nile tilapia culturing industry.

## 2. Materials and Methods

### 2.1. Ethics Statement

The protocol of animal husbandry and handling procedures used in this study followed the Animal Care Committee of Xiamen University.

### 2.2. Experimental Diet

Fishmeal, degossypolized cottonseed protein, soybean meal, and rapeseed meal were used as protein sources, while soybean oil was used as the main fat source to formulate the basic diet of Nile tilapia. Broad bean at an additive amount of 40% and 50% was also used as a plant-based protein source in the basic diet after balancing the content of lysine and methionine (the first and second limiting amino acids of tilapia), formulated as treatments G2 and G3. Then, based on G2 and G3, a crisping functional feed additive mixture (CFFAM, the formula is shown in Table 1) was added to prepare the other two crisping compound diets (G4 and G5). The basic diet without broad beans was used as the control (G1), and the commercial compound diet was the positive control (G6). After being crushed and sieved through an 80-mesh sieve, the raw materials were weighed according to the formula and gradually mixed. About 40% distilled water (*w*/*w*) was added to the mix and finally produced into a puffed diet by a heated extruder with a diameter of 3.0 mm. Diets were dried at 60 °C for 24 h and stored in the −20 °C refrigerator. The diet formula and proximate nutrient composition are shown in Table 1.

### 2.3. Experimental Fish and Feeding Management

Nile tilapia was provided by Guokeng Nile tilapia farm in Zhangzhou (Zhangzhou, China). The fish were all from the same batch of breeding with similar sizes were selected. After transportation to the cultured base, fish were fed with the G1 diet to acclimation for 7 days.

After acclimation, 360 fish with an initial body weight of 617.32 ± 1.64 g were randomly allocated to 24 barrels (diameter 1 m and water depth 1.3 m). Six quadruplicate diet groups (G1–G6) were set, with each replicate (barrel) containing fifteen fish. During the 90-day feeding trial, fish were fed twice daily (8:00 and 16:00) for about 1.0–1.5% of their body weight. The trial conditions were maintained as follows: water temperature of 25–28 °C, pH of 6–7.5, and dissolved oxygen ≥ 6.0 mg/L.

### 2.4. Sampling and Analyzing

#### 2.4.1. Proximate Composition Analysis

The proximate composition of the diets was analyzed according to the methods described by AOAC [17]. Crude protein content was determined based on the Kjeldahl method. The Soxhlet method was used to determine the crude lipid content.

#### 2.4.2. Sampling

At the end of the feeding trial, the final number of fish was counted for survival rate (SR) calculation. After being starved for 24 h, the fish were anesthetized with 45 mg/L eugenol (Shanghai Aladdin Biochemical Technology Co., Ltd., Shanghai, China) for later sampling. The body weight and length of all fish were measured.

Three fish per barrel were randomly selected for blood sample and dissection. Blood was taken from the tail vein using a 2.5 mL sterile syringe, left at 4 °C for 4 h, then separated by centrifugation for 10 min (4 °C, 3500× *g*). The serum was collected into a 200 μL sterile PCR tube and kept at −80 °C until analysis. The viscera and liver were dissected and weighed. Intestinal contents were collected in 1.5 mL sterile enzyme-free centrifuge tubes, frozen in liquid nitrogen, and quickly transferred to a −80 °C refrigerator. Anterior intestine tissue and muscle samples (10 mm × 10 mm × 15 mm) in the fixed position of the back were taken and fixed in 4% paraformaldehyde solution for histological observation. In addition, more muscle tissues (20 mm × 20 mm × 10 mm) were dissected to detect texture characteristics.

The weight gain rate (WGR), specific growth rate (SGR), feeding conversion ratio (FCR), condition factor (CF), viscera somatic index (VSI), and hepatosomatic index (HSI) were calculated using the following formulas:Weight gain rate (WGR, %) = (Wf − Wi)/Wi × 100
Specific growth rate (SGR, %/d) = (Ln (Wf) − Ln (Wi))/56 × 100
Survival rate (SR, %) = (Nf − Ni)/Ni × 100
Feeding conversion ratio (FCR) = Wd/(Wf − Wi)
Viscera somatic index (VSI, %) = Wv/Wf × 100
Hepatosomatic index (HSI, %) = WL/Wf × 100
Condition factor (CF, g/cm^3^) = Wf/(Lf3)
where Wi is the initial weight; Wf is the final weight; Ni is the initial number of fish; Nf is the final number of fish; Wd is the weight of digested food; Wv is the weight of viscera; WL is the weight of liver; and Lf is the final length.

#### 2.4.3. Detection of Serum Biochemical Parameters

The activity of total antioxidant capacity (T-AOC), superoxide dismutase (SOD), catalase (CAT), glutathione peroxidase (GSH-Px), adenosine deaminase (ADA), lysozyme (LZM), glutamate–pyruvate transaminase (GPT), and glutamic–oxaloacetic transaminase (GOT), as well as malondialdehyde (MDA) content were all measured using a commercial assay kit (Nanjing Jiancheng Biotechnology Research Institute, Nanjing, China) and performed in triplicate.

#### 2.4.4. Histological Observation

The muscle and intestine tissues were fixed in 4% paraformaldehyde solution for 24 h and then removed. After gradient dehydration, the fixed tissues were paraffin-embedded, sliced, and stained using the hematoxylin–eosin (H&E) staining method. The slices were imaged using an Eclipse Ci-L photomicroscope (Nikon, Shanghai, China). Muscle fibers of all taken images were measured using Image-Pro Plus 6.0 analysis software. Five muscle fibers of each slice were measured. Muscle fiber density was calculated via the total number of muscle fibers in each image and the area of the view:Muscle fiber density (n/mm^2^) = (total amount of muscle fibers)/(area of view)

#### 2.4.5. Muscle Texture Profile Analysis (TPA)

Texture profile analysis (TPA) of muscle samples (20 mm × 20 mm × 10 mm) was examined using a TA XT2i Texture Analyzer (Stable Micro Systems Ltd., Godalming, UK). A 35 mm diameter cylindrical probe (P/35R) was chosen for analysis. The test conditions involved two consecutive cycles of 30% compression with a 5.0 s interval between cycles. The pre-test speed, test speed, and post-test speed were all set at 1.0 mm/s. The measured parameters included hardness (g), fracturability (g), adhesiveness (g/s), springiness, cohesiveness, gumminess, chewiness (g), and resilience.

### 2.5. Intestinal Microbiome Analysis

Intestinal content samples were conducted with total DNA extraction according to the CTAB method, and 1% agarose gel electrophoresis was used to detect the quality. The genomic DNA was amplified through PCR using the specific primers with a barcode (Appendix A). PCR products were detected by 1% agarose gel electrophoresis and purified with Agencourt AMPure XP nucleic acid purification kit (Beckman, Brea, CA, USA). Purified PCR products were used for gene library construction, and finally, a high-throughput sequence was performed with Illumina MiSeq platform (Illumina, San Diego, CA, USA).

The raw data (Fastq data) were quality-controlled using Trimmomatic (v0.36) and Pear (v0.9.6) software to obtain clean data. All effective data were clustered using the UPARSE method by Qiime (v1.8.0) and vsearch (2.7.1) software, and species annotation was performed through the RDP classifier algorithm based on the Silva138 database. Qiime software (v1.8.0) was used to determine the alpha diversity indices (PD whole tree, Shannon, Simpson, and Chao1). R software (v4.3.1) was adopted to determine the beta diversity through the principal component analysis (PCA). Further, the LEfSe package in Python was utilized for linear discriminant analysis of the effect size (LEfSe), and functional annotation information was obtained using PICRUSt (v2.0) according to the KEGG database.

### 2.6. Statistical Analysis

The analysis of growth performance, serum biochemical, TPA, muscle, and intestinal tissue morphology parameters analyses were conducted using SPSS 22.0 (SPSS, Chicago, IL, USA). The results were presented as mean ± standard deviation (SD). To investigate differences among groups, Shapiro–Wilk and Levene’s test was employed to make the normality test and homogeneity of variance test, then a one-way analysis of variance (ANOVA) was conducted. When there were statistically significant differences, Duncan’s multiple-range test was used to compare the means of different groups. A *p*-value < 0.05 was recognized as statistically significant.

## 3. Results

### 3.1. Growth Performance

The effects of different diets on the growth performance of Nile tilapia are shown in Table 2. There was no significant difference in survival rate among treatments (*p* > 0.05, same as below), although the survival of Nile tilapia was the highest in treatments G2 and G5 (both at 87.50 ± 6.84%), the lowest in G6 (76.19 ± 17.98%). The FBW, WGR, and SGR of the G2, G3, G4, and G5 groups were significantly higher than those of the G1 and G6 (*p* < 0.05, same as below). Correspondingly, the FCR of treatments G2, G3, G4, and G5 were significantly lower than those of G1 and G6. In particular, the FCR of G3 was the lowest at 2.53, while the highest FCR was obtained from G1 and G6 (3.87 and 3.68, respectively). Significantly low VSIs were found in treatments G1 (5.71 ± 0.76%) and G2 (6.04 ± 1.23%); however, G6 treatment had the highest VSI at 6.93 ± 1.81%. There was no significant difference between the treatments in CF and HSI of Nile tilapia.

Table 3 summarizes the results of serum biochemical parameters of Nile tilapia after being fed with different diets. Fish from treatment G1 presented the highest serum MDA content at 9.70 nmol/mL, while those from G2 and G4 had the lowest levels at 7.50 and 7.49 nmol/mL, respectively. However, no statistical difference in serum MDA was detected between treatments. Additionally, the G1 treatment presented the highest T-AOC, ADA, and GOT levels in fish serum, and its total antioxidant capacity was significantly higher than in other treatments. The SOD in the serum of tilapia in the G3 group was significantly higher than that in the G2 and G4 groups, while the SOD in the serum of tilapia in the G3 group was significantly lower than that in the G1 group. The CAT activity in the serum of tilapia in the G3 had a significantly lower value than that in the G1 group, while the CAT activity in the serum in the G4 group was significantly higher than that in the G1 group. The GSH-Px activity in the serum of tilapia in the G3, G4, and G5 groups was significantly higher than that in the G1 group. There was no significant difference in the activity of LZM and GPT in fish serum between groups.

### 3.2. Characteristics of Muscle Fibers

Figure 1 and Figure 2 show the muscle H&E staining results and the muscle fiber characteristics of Nile tilapia in each treatment. As shown in Figure 2A, there was no significant difference in the muscle fiber diameter of Nile tilapia among the treatments. However, the muscle fiber densities of tilapia in treatments G1, G3, G4, G5, and G6 were higher than that in G2 (99.02 fibers/mm^2^), with the highest muscle fiber density observed in treatment G6, which was 141.11 fibers/mm^2^ (Figure 2B).

### 3.3. Muscle Texture Profile

The results of TPA are shown in Table 4. The hardness, chewiness, and adhesiveness of fish muscle in treatment G5 were the highest at 1155.32 g, 659.61 mJ, and 696.65 g, respectively; however, significant differences were only detected between G5 and G1. Those in the G2, G3, G4, and G6 groups were also higher than those in the control but did not reach a significant difference. Nevertheless, there was no significant difference in the cohesiveness and resilience of Nile tilapia muscles among treatments.

### 3.4. Intestinal Histomorphological Parameters

The effects of different diets on the intestinal tissue structure of Nile tilapia are shown in Figure 3. From Figure 3, it could be observed that for the fish in all treatments, the villi of the anterior intestine cells were regular. Moreover, all tissue structures were intact under ×100 and ×400 magnification.

The changes in the intestinal histomorphology parameters of Nile tilapia in each treatment are shown in Table 5. There was no significant difference in the villus epithelial length, lamina propria thickness, villous epithelial length, and the muscular layer thickness of goblet cells per unit length of the anterior intestine of Nile tilapia across the treatments. The number of goblet cells per unit length of the intestine in the G2, G4, G5, and G6 groups was higher than that in the G1 group, but only the G6 group reached a statistically significant level.

### 3.5. Intestinal Microbiota

There were 147–415 OTUs (operational taxonomic units) clustered from sequences at a 97% similarity. The alpha diversity of the intestinal microbiota diversity and richness (Chao1, Shannon, Simpson, and PD whole-tree indices) of all treatments are shown in Figure 4. The results of the Chao1 index illustrated that the Chao1 indices of G2 and G3 were higher than those of other treatments, which means that the number of OTUs in their community was the highest. The Chao1 index is followed by treatments G5 and G4. A similar phenomenon also occurred in the PD whole-tree index. In addition, the Shannon and Simpson indices followed the same trend as the counterparts of the G2 and G3 groups, which were lower compared to the control, while the counterparts of the G4, G5, and G6 groups were higher compared to the G1 group.

The PCA (principal component analysis) of the intestinal microbiota of Nile tilapia is shown in Figure 5 to determine the beta diversity. The results showed that, according to the distance among the treatments, the intestinal microbiota of Nile tilapia presented two separate clusters. Concretely, G1, G2 and G3 formed one cluster, while G4, G5 and G6 formed another.

The relative abundance of the bacterial communities at the phylum level is presented in Figure 6. *Fusobacteriota*, *Firmicutes*, and *Bacteroidota* were the dominant microbes in all treatments. The relative abundance of *Bacteroidota* in G4, G5, and G6 treatments was higher than that in G1, G2, and G3 groups, while the relative abundance of other two dominant microbial phyla (*Fusobacteriota* and *Firmicutes*) in G4, G5, and G6 treatments was lower than that in G1, G2, and G3 groups. Moreover, an increase was also seen in the relative abundance of *Proteobacteria* in the G3, G4, and G5 groups compared to G1.

At the genus level, the relative abundance of the bacterial communities is illustrated in Figure 7. *Cetobacterium* was the most dominant microbial genus in all groups. However, the relative abundance of *Cetobacterium* was lower when the fish was fed with G4, G5, and G6 diets compared to G1, while the relative abundance of *Bacteroides* and *Macellibacteroides* was higher. In addition, there was a lower value in the relative abundance of *Plesiomonas* in the G4 and G5 compared to other treatments.

LefSe analysis was performed, and the results are shown in Appendix A. The results showed differences in the abundance of 58 intestinal microbiota species. The main different bacteria in the fish intestine of treatment G1 were *Bacillaceae*, *Dietziaceae*, and *Staphylococcaceae*. The main different bacteria in the fish intestines of treatment G2 was the Corynebacterium family; the main bacterial groups in the fish intestines of treatment G3 were *Vibrionaceae* and *Chthoniobacteraceae*. There was no significant difference in the abundance of intestinal microbiota between G4 and other treatments. There were four microorganisms with significant differences in the treatment G5, among which the order *Bacteroidales* was the predominating microbiota. Eleven microorganisms with significant differences were found in the treatment G6, with the families *Bacteroidaceae* and *Tannerella* as the predominating microbiota.

The functional prediction analysis of differential microbiota showed no significant impact on the first level of the KEGG pathway between treatments. Differences in metabolism function on the second level of the KEGG pathway are shown in Appendix A. In the metabolism of co-factors and vitamins, energy metabolism, and the amino acid metabolism pathways, the G4, G5, and G6 groups were significantly higher than the G1, G2, and G3 groups. In the carbohydrate metabolism and glycan biosynthesis metabolism pathways, all crisping groups (G2–G6) were significantly higher than the control (G1), and treatments G4 and G6 were higher compared to others.

## 4. Discussion

### 4.1. Effects of Broad Bean Diets on the Growth of Nile Tilapia

Owing to their high nutritional value, ease of preservation, and low production cost, broad beans (*Vicia faba* L.) are a rich source of plant protein, nutrients, and dietary fiber to animals and humans [18,19,20]. As one of the cheapest protein sources, broad beans have been widely used in animal diets [21,22]. However, legumes have the disadvantage of lacking sulfur-containing amino acids, especially cysteine and methionine, that will affect the biological value and restrict the usage of broad beans [23]. In addition, the presence of antinutritional factors (including trypsin inhibitors, condensed tannins, phytic acid, saponins, lectins, and favism-inducing factors) also limits the use of legumes in aquaculture [24]. The presence of antinutritional factors in broad beans may hinder protein digestibility, causing a reduction in protein bioavailability [25]. As a consequence of the antinutritional factors, it likely leads to the poor absorption of some essential amino acids and results in growth depression in cultured animals [26,27].

In this study, there was no adverse effect on the growth performance of Nile tilapia fed with broad bean diets. On the contrary, feeding crisping diets did improve the WGR, SGR, and FCR of fish. However, previous study has shown that a high broad bean content (75% and 100%) led to a significant deterioration in fish growth performance [16], which may be because broad beans did not replace the added amount of fishmeal but rather replaced a portion of the plant-sourced protein in this study. Also, this difference could be caused by the age/size of the fish or the duration of this study.

### 4.2. Effects of Broad Bean Diets on the Serum Indices of Nile Tilapia

Serum biochemical indices are related to the fish species and living environment, nutritional status, and also reflect the health status of the fish [28]. The antioxidant enzymes in the natural antioxidant defense system of organisms can eliminate peroxides [29]. The most important antioxidant enzymes include superoxide dismutase (SOD), glutathione peroxidase (GSH-Px), and catalase (CAT) [30]. SOD disproportionates superoxide anions into hydrogen peroxide, while CAT and GSH-Px can reduce hydrogen peroxide, thereby preventing the production of highly toxic hydroxyl radicals [31]. Importantly, GSH-Px can also reduce the hydroperoxides of polyunsaturated fatty acids, thereby offsetting the toxic effects of lipid peroxidation [32]. At the same time, because lipid peroxidation is initiated by free radicals, unsaturated fatty acids form conjugated dienes in the initial stage of lipid oxidation caused by free radicals and some oxidation, then form cyclic peroxides, and the final main product is malondialdehyde (MDA) [33]. In addition, total antioxidant capacity (T-AOC) refers to the total antioxidant level composed of various antioxidant substances and antioxidant enzymes, which can be directly used to evaluate antioxidant capacity [34].

In this study, the T-AOC levels of all broad bean diet groups were significantly lower than that of the control, meaning that the supplementation of broad beans weakened the antioxidant capacity of fish; however, there was no difference between treatments with different additive amounts. Moreover, even with additional crisping functional packages, the side effects on the antioxidant capacity of the antinutritional factors could not be eliminated. As for antioxidant enzymes, four broad bean replacement treatments showed increased SOD, CAT, and GSH-Px activities. After replacing a portion of plant protein with broad beans in diets, in the case of reduced antioxidant capacity, tilapia received oxidative stress from broad beans, increasing antioxidant enzyme activity. However, free radicals did not cause lipid peroxidation according to non-increased peroxide product MDA.

Glutamate–pyruvate transaminase (GPT), glutamic–oxaloacetic transaminase (GOT), and adenosine deaminase (ADA) levels in the serum are indices related to the liver health of fish. Typically, GPT and GOT exist in the liver. When liver disease occurs, the permeability of liver cells increases, and GPT and GOT are released into the blood, leading to increased activity of GPT and GOT in the serum [35]. ADA is an important enzyme system for the metabolism and breakdown of adenosine in cells. The activity of ADA in serum often increases when liver cells are damaged and is more sensitive than GPT in reflecting chronic liver cell damage [36]. In this study, only the ADA and GOT had the variation compared to the control, and this change is positive. Therefore, the broad bean diets did not cause damage to the liver of the experimental fish.

Lysozymes (LZMs) are a part of the immune system, and their activity can reflect the immune capacity of an organism [37]. According to the results of LZM in this study, there was seldom an influence on immune capacity made by adding broad beans. In a study on beluga (*Huso huso*), broad bean meal can be used successfully in a diet of up to 10% without adverse effects on serum biochemical parameters [26]. However, Gan et al. (2017) indicated that the anti-oxidative activity decreased with the increasing broad beans level in grass carp [27]. Different fish species have varying degrees of response to broad beans, and the additional amount designed in this experiment did not reach the level where change can occur.

### 4.3. Effects of Broad Bean Diets on the Flesh Quality of Nile Tilapia

Muscle hardness, springiness, adhesiveness, and chewiness directly reflect the meat quality, while hardness and chewiness are key parameters that directly reflect the degree of fish fragility [38]. Muscle fibers are the basic units of muscle tissue, and their density and diameter directly affect the meat quality [39]. Research on grass carp [40], Yellow River carp [41], and Atlantic salmon [42] have indicated that muscle fiber density is negatively correlated with muscle fiber diameter but positively correlated with muscle tissue hardness and chewiness. In other words, the smaller the diameter of the muscle fiber would result in an increased number of muscle fibers per unit area—that is, the greater the density of the muscle fiber, and ultimately improve the hardness and chewiness of the meat and achieve a ‘crisping’ effect. In this trial, the muscle hardness, adhesiveness, and chewiness of all broad bean diets increased compared to the basic diet (G1). Increasing the amount of addition can increase these flesh quality parameters, and adding a crisping functional package to the diet can also improve muscle hardness, adhesiveness, and chewiness of tilapia. In addition, the effect of the crisping formula used in this study was superior to the commercial diet (G6), especially treatment G5, which has the potential to be developed as a new type of tilapia crisping-specific diet.

### 4.4. Effects of Broad Bean Diets on the Intestine of Tilapia

Previous research has shown that the digestive and absorption capacity of the intestine is closely related to parameters such as intestinal villus length, muscle layer thickness, and lamina propria thickness [43]. The results of the present experiment indicate that feeding Nile tilapia with the formula feed containing broad beans did not significantly affect its foregut tissue morphology. However, feeding a formula feed containing broad beans and a crisping functional package increased the number of goblet cells per unit length in the intestine of Nile tilapia. Goblet cells are a typical type of mucus cell associated with the intestinal mucosal immune system and have a protective effect on the intestinal tissue structure [44]. They can resist infection, regulate epithelial cell integrity, and respond to foreign antigens [45]. In addition, goblet cells can synthesize secreted mucoprotein (MUC2) and bioactive molecules, reducing bacterial adhesion to the intestinal wall and intestinal permeability in the intestine. The experiment found that adding broad beans to grass carp feed significantly induced intestinal villi shedding and fusion, increased lamina propria, and decreased eosinophilic granulocytes, destroying intestinal structure [40]. After feeding Nile tilapia with broad beans, it was found that the intestinal villi were slightly damaged, and the muscle thickness of the intestine was increased, suggesting modest intestinal injury [6]. The damage caused by broad beans added diet to fish in the research above was also found in the comparison between the G1 and G3 treatments, which may be due to the lack of a crisping functional package. The supplementation of broad beans will induce inflammation of the intestine and finally result in intestinal injuries (cells and structure) [46]. A recent study has shown that adding short-chain fatty acids to the crisping functional package protects the integrity of the intestinal mucosal barrier and regulates the intestinal immune system [47]. All these results indicated that the crisping functional package can be applied to tilapia crisping formula feed, which helps to improve the integrity of the intestinal tissue structure of Nile tilapia and the health of the fish intestine.

The intestinal microbiota of fish is closely linked to the health of their host organisms. Normal intestinal microbiota in fish guts is conducive to the digestion and absorption of nutrients, and it also helps to regulate the immune response [48]. The composition of fish intestinal microbiota could be easily affected by food and feeding methods, and its composition significantly affects the health of the fish gut [49]. In this study, Nile tilapia fed with broad beans without a crisping functional package had a decrease in the biological diversity of intestinal microbiota. However, a higher α diversity was observed when the crisping functional package was applied. The higher the degree of α diversity, the more stable the composition of intestinal microbiota, the stronger the ability to resist external influences, adaption, and self-balance recovery, and is more beneficial to the health of the host [50]. The results showed that the additional broad beans to Nile tilapia might destroy its intestinal microbiota composition, while the addition of functional packages on its basis can alleviate the destruction of its intestinal microbiota composition. Similar results were reported in grass carp, as their intestinal microbiota were altered when broad beans were applied as feed supplementation [51,52]. In the case of the yellow catfish (*Tachysurus fulvidraco*), using broad beans in feed reduced the abundance and diversity of bacteria in the gut to varying degrees, causing a significant decrease in the rate of weight gain [53].

In addition, the predominant bacterial species in the digestive tract can be affected by feed supplements, and they largely determine the function of the fish gut microbiota community [54,55,56]. In this study, the fish intestines from the six treatments were enriched with *Bacillus*, *Corynebacterium*, *Vibriaceae*, *Bacteroides*, and *Bacteroidaceae*, respectively. Studies have proved that the colonization of *Bacillus* in the intestines of animals can effectively protect their intestinal tissues and improve intestinal health [57]. A species of the *Vibrio* with a flagellum on the tail is a type of anaerobic Gram-negative bacteria, and some species, such as *Vibrio cholerae*, *Vibrio parahaemolyticus*, and *Vibrio harveyi*, are pathogenic bacteria of humans or aquatic products. *Bacteroides* are Gram-stain-negative bacteria, peptones, or metabolic intermediates that assist digestion in the intestines of aquatic animals [58]. The change in the relative abundance of the aforementioned bacterial species will finally result in the variation in intestinal health. Such a result is quite similar to previous studies. For example, a study found that the use of in vitro fermented broad beans can significantly increase the abundance of beneficial bacteria, such as *Bifidobacterium* sp. and *Enterococcus* (Lactobacillus), and achieve the function of optimizing host intestinal health [59]. Moreover, grass carp fed with broad bean hydrolysate reportedly improved intestinal microbiota composition and reduced pathogenic bacteria abundance [60].

The intestinal microbiota is an essential part of the ecosystem in the organism, and changes in the intestinal microbiota impact the function of the intestinal microbiota [61]. Intestinal dysbacteriosis impacts metabolism in organisms, including but not limited to lipid metabolism, sugar metabolism, and amino acid metabolism. The accumulation and imbalance of these metabolites are also closely related to the occurrence and progression of various diseases [62]. According to our study, in the metabolism of co-factors and vitamins, energy metabolism, and the amino acid metabolism pathways, treatments G4–G6 were significantly higher than treatments G1–G3. These results suggested that using a crisping functional package may affect the metabolism of co-factors, vitamins, amino acids, and energy. In other words, such results indicated that adding functional packs to tilapia crisping feed could improve intestinal microbiota function and promote the metabolic processes of fish. On the contrary, some studies reported that broad beans could cause damage to the function of the intestinal microbiota. A typical example is that when weaned piglets are fed compound feed with more than 20% broad bean content, the amino acid metabolism function in predicting intestinal microbiota community function is negatively affected, indicating that their amino acid metabolism process is disordered [63].

The main components of the crisping functional package are short-chain fatty acids, antimicrobial peptides, bile acids, plant extracts, and taurine, among which short-chain fatty acids, antimicrobial peptides, and bile acids have positive effects on the structure and function of intestinal microbiota. Short-chain fatty acids are produced by the metabolism of the microbiota in the gastrointestinal tract and have a variety of physiological functions on intestinal epithelial cells, such as increasing metabolic levels and reducing intestinal inflammation [64]. Antimicrobial peptides are natural antimicrobials produced by intestinal epithelial cells that recognize and destroy pathogenic microorganisms and prevent intestinal infections [65]. Bile acids are compounds synthesized by the liver and secreted into the intestine, which are involved in digestion and lipid metabolism and can maintain the balance of the intestinal microbiota by interacting with the intestinal microbiota [66]. These three bioactive substances work together to maintain the diversity and stability of the intestinal microbiota, promote the integrity of the intestinal mucosal barrier, reduce the intestinal inflammatory response, and benefit intestinal health. All methods above aim at flesh quality improvement through nutritional approaches.

## 5. Conclusions

In brief, broad bean diets would cause no adverse effect on the growth performance of Nile tilapia. The supplementation of broad beans weakened the antioxidant capacity of fish but had no negative influence on liver health and the immune system. Increasing the amount of addition can increase muscle quality values, and the crisping functional package added to the diet can also improve muscle hardness, adhesiveness, and chewiness. In addition, the crisping functional package helped to improve the integrity of the intestinal tissue structure of Nile tilapia and the optimization of the fish intestinal microbiota. However, it should be pointed out that there are still some limitations in our study, like the short feeding time and the big initial fish weight; further trials can be conducted to perfect this study.

## Figures and Tables

**Figure 1 animals-13-03705-f001:**
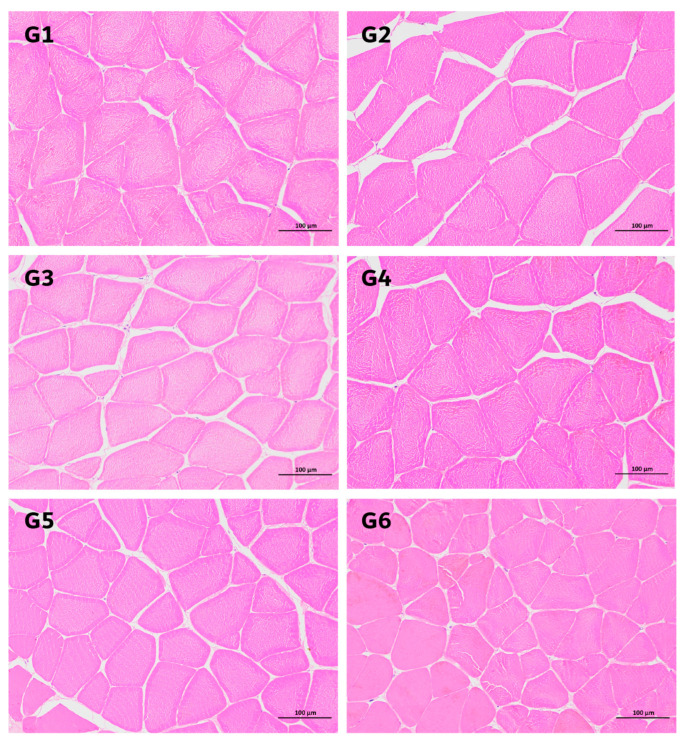
Histological section of muscle fibers of Nile tilapia (*Oreochromis niloticus*) fed different experimental diets (200×). For the formulation and nutrient composition of experimental diets G1–G6, please refer to Table 1.

**Figure 2 animals-13-03705-f002:**
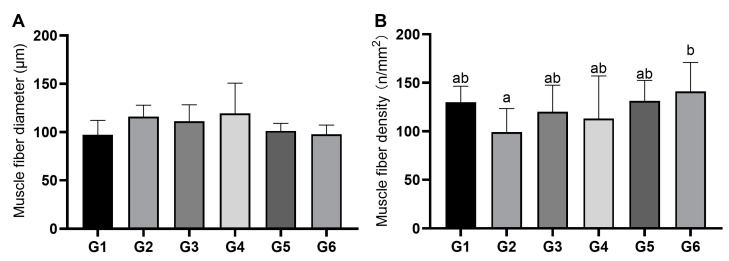
The diameter and density of muscle fibers of Nile tilapia (*Oreochromis niloticus*) fed with different experimental diets. Means with different letters suggest significant differences (*p* < 0.05). For the formulation and nutrient composition of experimental diets G1–G6, please refer to Table 1. (**A**) Muscle fiber diameter and (**B**) muscle fiber density.

**Figure 3 animals-13-03705-f003:**
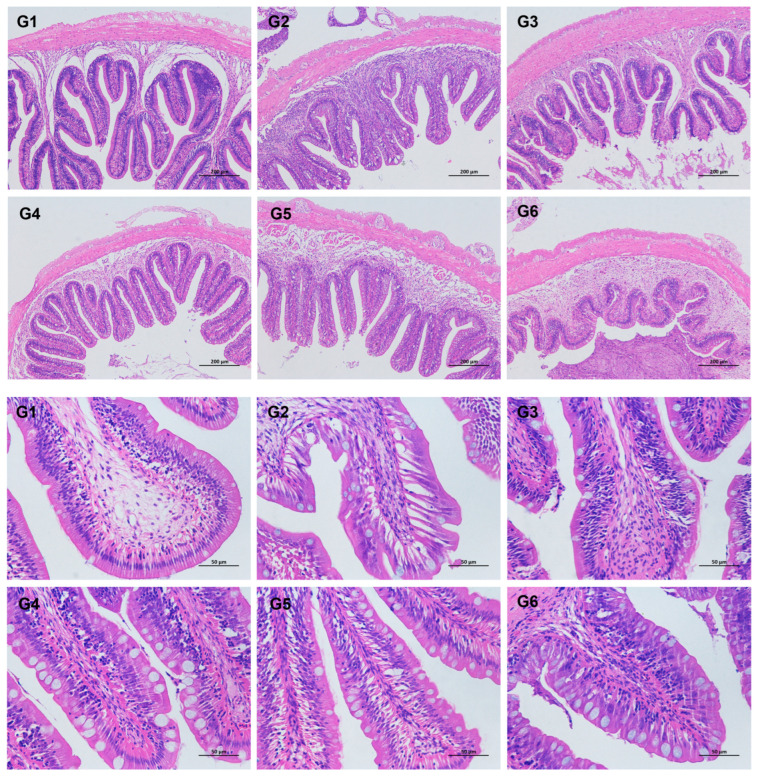
Intestine morphology of Nile tilapia (*Oreochromis niloticus*) fed different experimental diets: top 6 images (HE, 100×); last 6 images (HE, 400×). For the formulation and nutrient composition of experimental diets G1–G6, please refer to Table 1.

**Figure 4 animals-13-03705-f004:**
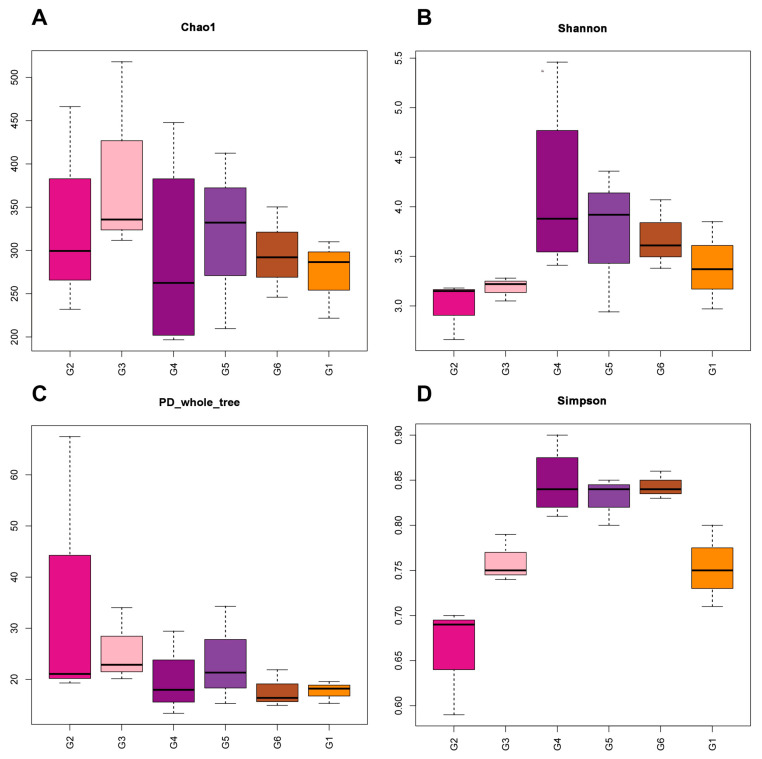
Analysis of alpha diversity indices for Nile tilapia (*Oreochromis niloticus*) fed different experimental diets. For the formulation and nutrient composition of experimental diets G1–G6, please refer to Table 1. (**A**) Chao1; (**B**) Shannon; (**C**) PD whole tree; and (**D**) Simpson.

**Figure 5 animals-13-03705-f005:**
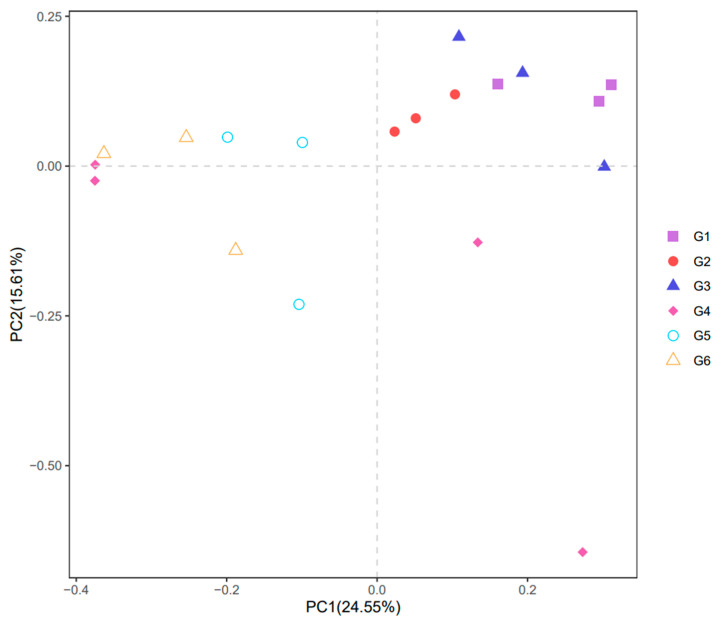
PCA analysis of the intestinal microbiota of Nile tilapia fed different experimental diets. For the formulation and nutrient composition of experimental diets G1–G6, please refer to Table 1.

**Figure 6 animals-13-03705-f006:**
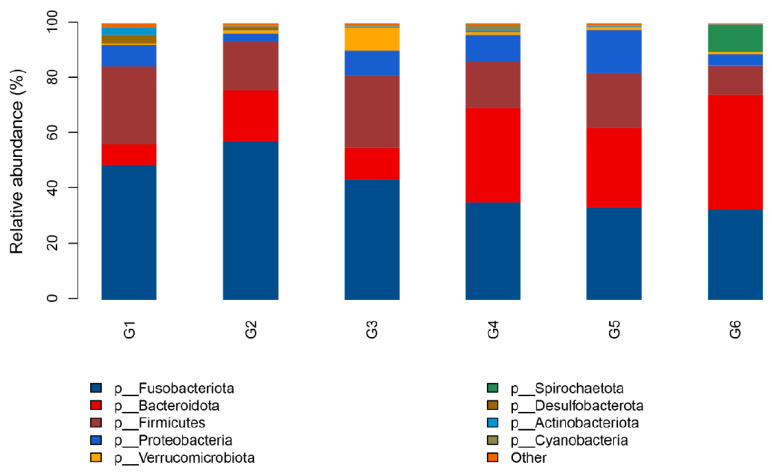
Relative abundance of the intestinal microbiota at the phylum level of Nile tilapia (*Oreochromis niloticus*) fed different experimental diets. For the formulation and nutrient composition of experimental diets G1–G6, please refer to Table 1.

**Figure 7 animals-13-03705-f007:**
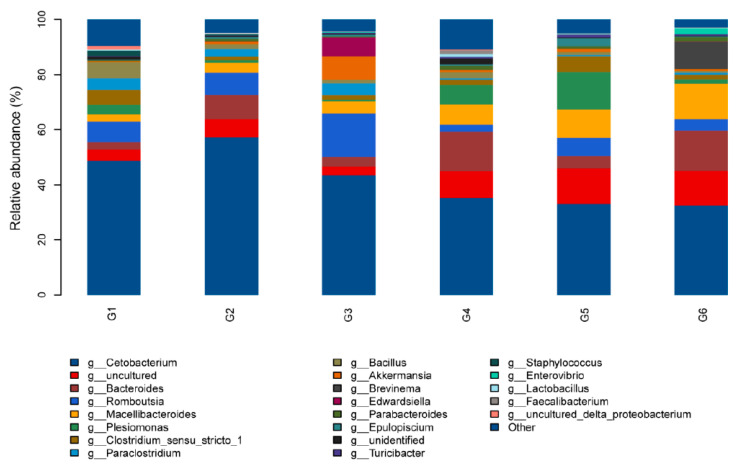
Relative abundance of the intestinal microbiota at the genus level of Nile tilapia (*Oreochromis niloticus*) fed different experimental diets. For the formulation and nutrient composition of experimental diets G1–G6, please refer to Table 1.

**Table 1 animals-13-03705-t001:** Formulation and nutrient composition of the experimental diets (% dry weight).

	Experimental Diets (Treatments)					
	G1	G2	G3	G4	G5	^1^ G6
**Ingredient (%)**						
Fish meal	10	10	10	10	10	
Degossypolized cottonseed protein	3	8	16.3	8	16.3	
Soybean meal	26.5	18.1	0	18.1	0	
Rapeseed meal	25.5	0	0	0	0	
Rice bran	10	0	0	0	0	
Broad bean	0	40	50	40	50	
Soybean oil	4.4	3.5	3.2	3.5	3.2	
Wheat flour	18	17.6	17.5	17.6	17.5	
^2^ Permix	2.6	2.6	2.6	2.6	2.6	
Lysine	0	0.07	0.19	0.07	0.19	
L-methionine	0	0.19	0.23	0.19	0.23	
^3^ CFFAM				0.5	0.5	
**Proximate composition (%)**						
Crude lipid	6.39	6.32	6.32	6.39	6.39	6.1
Crude protein	32.79	32.75	32.75	32.79	32.79	32.23
Lysine	1.90	1.90	1.90	1.90	1.90	1.86
Methionine	0.56	0.56	0.56	0.56	0.56	0.52

^1^ G6 diet: A commercial crisping compound diet, so its formula is not listed. The nutrient content is the measured value. ^2^ Permix: a 1% freshwater fish general purpose permix (Biotechnology Group Co., Ltd., Suzhou, China) containing 1.5% calcium dihydrogen phosphate and 0.1% choline. ^3^ CFFAM: mainly included Macleaya cordata extract (5%), hydrolyzed tannic acid (10%), short-chain fatty acids (20%), bile acid (20%), antibacterial peptide (20%), taurine (10%), and other carrier substances (15%).

**Table 2 animals-13-03705-t002:** Growth performance and morphological indices of Nile tilapia (*Oreochromis niloticus*) fed different experimental diets.

Index	Experimental Diets					
	G1	G2	G3	G4	G5	G6
IBW (g)	616.44 ± 1.50	615.83 ± 0.64	617.25 ± 2.69	615.75 ± 2.33	619.42 ± 3.59	619.22 ± 4.55
FBW (g)	717.77 ± 24.75 ^b^	758.79 ± 17.06 ^a^	783.27 ± 31.06 ^a^	774.77 ± 34.30 ^a^	774.75 ± 10.70 ^a^	729.37 ± 15.06 ^b^
WGR (%)	16.43 ± 3.82 ^b^	23.22 ± 2.84 ^ab^	26.89 ± 4.66 ^a^	25.84 ± 5.89 ^a^	25.08 ± 1.28 ^a^	17.8 ± 2.87 ^b^
SGR (%/day)	0.17 ± 0.04 ^b^	0.23 ± 0.02 ^ab^	0.26 ± 0.04 ^a^	0.25 ± 0.05 ^a^	0.25 ± 0.01 ^a^	0.18 ± 0.03 ^b^
FCR	3.87 ± 0.99 ^a^	2.73 ± 0.36 ^b^	2.53 ± 0.31 ^b^	2.53 ± 0.47 ^b^	2.69 ± 0.27 ^b^	3.68 ± 0.21 ^a^
SR (%)	80.95 ± 10.91	87.50 ± 6.84	85.71 ± 5.83	80.36 ± 14.72	87.50 ± 6.84	76.19 ± 17.98
HSI (%)	1.31 ± 0.39	1.39 ± 0.40	1.37 ± 0.36	1.41 ± 0.42	1.47 ± 0.41	1.29 ± 0.45
VSI (%)	5.71 ± 0.76 ^b^	6.04 ± 1.23 ^b^	6.20 ± 1.27 ^a^	6.11 ± 1.22 ^a^	6.28 ± 1.24 ^a^	6.93 ± 1.81 ^a^
CF (g/cm^3^)	3.60 ± 0.29	3.55 ± 0.23	3.55 ± 0.28	3.48 ± 0.30	3.46 ± 0.71	3.58 ± 0.31

Data are expressed as mean ± standard deviation. Values in the same line with different letters suggest significant differences (*p* < 0.05). IBW—initial body weight; FBW—final body weight; WGR—weight gain rate; SGR—specific growth rate; FCR—feeding conversion ratio; SR—survival rate; CF—condition factor; VSI—viscera somatic index; HSI—hepatosomatic index.

**Table 3 animals-13-03705-t003:** Serum biochemical indices of Nile tilapia (*Oreochromis niloticus*) fed different experimental diets.

Index	Experimental Diets					
	G1	G2	G3	G4	G5	G6
MDA (nmol/mL)	9.70 ± 2.18	7.50 ± 0.32	8.30 ± 2.24	7.49 ± 1.24	8.01 ± 3.10	9.07 ± 2.76
T-AOC (U/mL)	1.10 ± 0.1 5 ^a^	0.86 ± 0.09 ^ab^	0.78 ± 0.10 ^b^	0.76 ± 0.08 ^b^	0.76 ± 0.24 ^b^	0.84 ± 0.23 ^ab^
SOD (mmol/L)	16.50 ± 0.08 ^a^	18.34 ± 0.12 ^b^	13.62 ± 0.71 ^c^	20.98 ± 0.61 ^d^	15.85 ± 0.33 ^a^	20.29 ± 0.12 ^d^
CAT (mmol/L)	68.72 ± 5.06 ^b^	61.39 ± 6.32 ^b^	28.26 ± 4.29 ^c^	152.00 ± 28.11 ^a^	75.52 ± 21.24 ^b^	157.91 ± 3.41 ^a^
GSH-Px (mmol/L)	370.55 ± 16.83 ^cd^	382.92 ± 11.35 ^c^	578.96 ± 27.44 ^a^	561.45 ± 11.64 ^a^	459.43 ± 27.18 ^b^	345.26 ± 19.91 ^d^
ADA (U/L)	3.61 ± 0.48 ^a^	2.47 ± 0.38 ^b^	2.16 ± 0.69 ^b^	2.90 ± 0.68 ^ab^	2.42 ± 0.30 ^b^	2.17 ± 0.47 ^b^
LZM (U/mL)	280.00 ± 60.40	301.33 ± 12.22	270.00 ± 53.22	300.00 ± 35.48	316.00 ± 63.16	266.67 ± 44.06
GPT (U/L)	1.50 ± 0.35	1.66 ± 0.26	1.40 ± 0.12	1.83 ± 0.71	1.52 ± 0.50	1.70 ± 0.64
GOT (U/L)	4.25 ± 1.69 ^a^	1.73 ± 0.25 ^b^	2.26 ± 1.15 ^b^	1.66 ± 0.71 ^b^	1.88 ± 0.60 ^b^	1.51 ± 0.36 ^b^

Data are expressed as mean ± standard deviation. Values in the same line with different letters suggest significant differences (*p* < 0.05). MDA—malondialdehyde; T-AOC—total antioxidant capacity; SOD—superoxide dismutase; CAT—catalase; GSH-Px—glutathione peroxidase; ADA—adenosine deaminase; LZM—lysozyme; GPT—glutamate–pyruvate transaminase; GOT—glutamic –oxaloacetic transaminase.

**Table 4 animals-13-03705-t004:** Muscular texture characteristics of Nile tilapia (*Oreochromis niloticus*) fed different experimental diets.

Index	Experimental Diets					
	G1	G2	G3	G4	G5	G6
Hardness (g)	753.90 ± 20.04 ^b^	818.63 ± 130.95 ^ab^	1076.59 ± 368.57 ^ab^	933.93 ± 278.69 ^ab^	1155.32 ± 214.97 ^a^	982.84 ± 287.47 ^ab^
Springiness (mm)	0.95 ± 0.03	0.94 ± 0.02	0.93 ± 0.02	0.92 ± 0.03	0.95 ± 0.01	0.94 ± 0.01
Adhesiveness (g)	428.12 ± 33.57 ^b^	495.44 ± 129.40 ^ab^	599.63 ± 212.53 ^ab^	516.86 ± 163.38 ^ab^	696.95 ± 138.91 ^a^	567.45 ± 115.57 ^ab^
Chewiness (mJ)	408.80 ± 30.37 ^b^	469.37 ± 128.85 ^ab^	557.56 ± 186.22 ^ab^	476.06 ± 145.10 ^ab^	659.61 ± 126.65 ^a^	536.55 ± 114.20 ^ab^
Cohesiveness	0.57 ± 0.04	0.61 ± 0.05	0.56 ± 0.03	0.55 ± 0.02	0.60 ± 0.01	0.59 ± 0.05
Resilience	0.71 ± 0.08	0.71 ± 0.04	0.74 ± 0.06	0.74 ± 0.03	0.66 ± 0.05	0.73 ± 0.12

Data are expressed as mean ± standard deviation. Values in the same line with different letters suggest significant differences (*p* < 0.05).

**Table 5 animals-13-03705-t005:** Histomorphological parameters of the foregut of Nile tilapia (*Oreochromis niloticus*) fed different experimental diets.

Index	Experimental Diets					
	G1	G2	G3	G4	G5	G6
Villus epithelial length (mm)	0.4 ± 0.08	0.29 ± 0.02	0.36 ± 0.07	0.29 ± 0.02	0.29 ± 0.04	0.3 ± 0.08
Lamina propria thickness (mm)	0.51 ± 0.07	0.4 ± 0.03	0.46 ± 0.07	0.4 ± 0.02	0.42 ± 0.09	0.42 ± 0.06
Villous epithelial length (mm)	0.73 ± 0.12	0.42 ± 0.09	0.66 ± 0.13	0.56 ± 0.05	0.52 ± 0.04	0.56 ± 0.13
Muscular thickness (mm)	0.12 ± 0.01	0.12 ± 0	0.13 ± 0.02	0.08 ± 0.01	0.12 ± 0.02	0.12 ± 0.02
Number of goblet cells per unit length (ind./mm)	23.76 ± 3.52 ^ab^	27.46 ± 1.43 ^ab^	20.93 ± 8.08 ^b^	28.22 ± 9.91 ^ab^	41.69 ± 5.17 ^ab^	57.73 ± 8.23 ^a^

Data are expressed as mean ± standard deviation. Values in the same line with different letters suggest significant differences (*p* < 0.05).

## Data Availability

The data presented in this study are available on request from the corresponding author.

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
