# Peer review of "Effects of Broad Bean Diet on the Growth Performance, Muscle Characteristics, Antioxidant Capacity, and Intestinal Health of Nile Tilapia (Oreochromis niloticus)"

_animals, 2023, doi:10.3390/ani13233705_

Round 1
Reviewer 1 Report
Comments and Suggestions for Authors
This study investigated the use of broad bean in the formulated feed of Nile tilapia in terms of growth performance, muscle characteristics, antioxidant capacity, and intestinal health. The subject of this paper is important in aquaculture practice. However, some major questions/defects should be addressed before accepted:
1. To my knowledge tilapia can reach a weight of about 1 kg. Why the weight gain was so low after a 90-d feeding? Please give a reasonable explanation.
2. Abstract & Conclusion: and a lower FCR was obtained.
3. Table 2, HSI
4. Please mark the group name of each picture
5. Figure 2B, Please check the significance level of observed difference among groups because of the large error bar with G4 group.
6. Figure 3, Please present the same position, orientation, and viewing angle per micrograph.
7. P-values should be italic.
8. Table 5, villus epithelial length
9. Bacteria at genus level should be italic.
10. Where is Figure 9 ???
11. I think there are too many figures about the intestinal microbiota, some figures could be deleted or added to the additional file.
12. The statements in Conclusion and Abstract are repeated several times, these have to be modified.
13. I wonder how many fish finally became the crisped fish in each group? And how much is the cost of the diet G5, comparing to commercial diet G6? If the results are good, it could be a successful new commercial fumular.
Comments on the Quality of English LanguageMinor editing of English language required
Author Response
- To my knowledge tilapia can reach a weight of about 1 kg. Why the weight gain was so low after a 90-d feeding? Please give a reasonable explanation.
Reply: There are main two reasons: first the initial weight of the fish in this study are larger than that in other researches, weight gain rate will slow down as the age of fish grow. Second, The optimal water temperature for Nile tilapia is around 30℃, our experiment was performed at Autumn season, the water temperature range was 25~28°C (described in line 117). A low temperature will lead to a low growth rate.
Reference:
Qiang, J., Yang, H., Wang, H., Kpundeh, M.D. and Xu, P., 2012. Growth and IGF-I response of juvenile Nile tilapia (Oreochromis niloticus) to changes in water temperature and dietary protein level. Journal of Thermal Biology, 37(8), pp.686-695.
Xie, S., Zheng, K., Chen, J., Zhang, Z., Zhu, X. and Yang, Y., 2011. Effect of water temperature on energy budget of Nile tilapia, Oreochromis niloticus. Aquaculture Nutrition, 17(3), pp.e683-e690.
- Abstract & Conclusion: and a lower FCR was obtained.
Reply: The mistake has been revised.
- Table 2, HIS
Reply: The mistake has been revised.
- Please mark the group name of each picture
Reply: the group name of each picture Figure 1 and Figure 3 has been added.
- Figure 2B, Please check the significance level of observed difference among groups because of the large error bar with G4 group.
Reply: The one-way analysis of variance (ANOVA) was conducted again, there are still no difference among groups. (F = 0.9463, p = 0.4864)
- Figure 3, Please present the same position, orientation, and viewing angle per micrograph.
Reply: The figure 3 has been revised.
- P-values should be italic.
Reply: P-values have been italic in submitted manuscript version It may be led by manuscript editing and typesetting mistakes. We have revised all the problems like this.
- Table 5, villus epithelial length
Reply: Thank you for your advice, the expression has been improved in Table 5.
- Bacteria at genus level should be italic.
Reply: Bacteria at genus level has been italic in submitted manuscript version. It may be led by manuscript editing and typesetting mistakes. We have revised all the problems like this.
- Where is Figure 9 ???
Reply: The manuscript we submitted contains Figure 9 after check. It may be led by manuscript editing and typesetting mistakes. We have added the Figure 9 [S2 in supplemental file] again.
- I think there are too many figures about the intestinal microbiota, some figures could be deleted or added to the additional file.
Reply: Thank you for your advice, Figure8 and Figure 9 have been moved to the additional file.
- The statements in Conclusion and Abstract are repeated several times, these have to be modified.
Reply: The expression in Conclusion has been modified [Line 563-571].
- I wonder how many fish finally became the crisped fish in each group? And how much is the cost of the diet G5, comparing to commercial diet G6? If the results are good, it could be a successful new commercial fumular.
Reply: Thank you so much for recognizing our research. The fish meat has become crispy based on the results in Table 4. We are so sorry that the rate of crisped fish was not counted, but the proportion should be high according to the diner tasting. We will conduct statistics in the future. In addition, according to the calculation results, the cost of the diet G5 is 5850 RMB/ton while the cost of the diet G6 is 6800 RMB/ton. Our designed crisping diet G5 is about 1000 RMB cheaper per ton than commercial crisping feed.
Reviewer 2 Report
Comments and Suggestions for Authors
This manuscript represents a lot of work. There are a number of issues that need to be addressed to improve the paper.
The wording of the title needs to be improved. This is not "effects of the growth performance..." but "effects on the growth performance...". Also, in every mention of the scientific name of Nile tilapia (apart from in the references) the spelling is incorrect - it is Oreochromis not Orechromis.
In this manuscript the word "crisp" appears to have several meanings. It refers to a diet (line 12), a dietary supplement (line 19), a marketing description of a fish with particular flesh texture (line 24) and fish flesh properties (line 43). This is confusing and the authors need to use different words for the diet. It is also unclear whether crispy fish is eaten raw or if it needs to be cooked. This needs to be addressed.
There is no simple summary.
The Abstract needs to say what size of tilapia were used and how long the fish were fed on the diets. Also, it needs to be stated that whether the final weight of the tilapia is the typical size at which this species is harvested and marketed in the region. If it is not then then how appropriate is the study?
Lines 77 & 80 suggests the manuscript will deal with "breeding technique" and "breeding industry" but I did not read anything about this.
There is information missing about the manufacture of diets G1-G5. Line 96 talks about water being added and then a puffed diet is mentioned - but no indication of whether a cold compressed pellet or heated extruder was used.
Were the fish mixed sex or all male/all female?
The diets are described but there is nothing about the nature of diet used by the farm to grow the fish to 600g before the trial started. The fish were then fed diet G1 for 7 days and then most were changed to diets G2-G6 for the remainder of the trial. The fish were not sampled (other than weight/length) at the start of the trial so any and all suggestions that the values of various parameters increased or decreased in the various groups over the course of the trial are not correct.
How were the fish fed - to satiation? was uneaten food collected and weighed? There's no suggestion that feed allocation was measured until we get to FCR. How was water quality maintained - flow through? Recirculation? Aeration? What was the cause of mortality?
Line 137. Where is "the fixed position" that the muscle tissue was removed from? There is no information about when or how the muscle tissue was stored/tested for texture profiling. Was it done immediately, or after rigor mortis, at 4 degrees or room temperature, raw or after cooking?
Line 181. There needs to be a reference for the "specific primers with a barcode" used for the microbiome analysis.
Lines 195 and 199 are contradictory. Were data analysed by R or SPSS?
Figure 1. There is nothing on the photographs to show which image refers to which diet treatment. The font used for the scale bar is too small. It is clear that the muscle fibers are not circular in cross section, so how can they have a diameter? How was the 'diameter' of the 3, 4 or 5-sided cross sectional area calculated? In Figure 2 are the graphs (A) and (B) the correct way around? The scale bar on the photographs would suggest the muscle fibres are about 200um 'square' not 10um as the graphs and text (line 256) would suggest. Also, if you estimate the cross sectional area of a 10um x 10um fiber = 100um2 and then multiply this by 130 (the suggested number of fibers per mm2) you get 13,000um2. This is only 1.3% of the 1,000,000um2 that is 1 mm2. If you swap the graphs, then the muscle fiber is 200 x 200 = 40,000um2 and if there are 10 per mm2 then it is 400,000um2 which is 40% of the area. If I am correct then I have serious concerns about the credibility of this aspect of the work. Also, I cannot see how G4 muscle fiber density can be shown as 'b' when it clearly overlaps so much with G1, G3 and G4. Please re-check.
Figure 3 is not easy to look at. Please consider putting the six 100x images together and the six 400x images together rather than mixing them. The font for the scale bar is too small. Table 5 - why are all the values in mm when the um will give you more decimal places?
Lines 309 and 312 use "increased most" and "grew up(?)" when there was no time course for changes in microbiome. Line 340 suggests "was reduced after the fish were fed with G4-G6 for 90 days" but this is misleading since there was no measurement made at day 0. Similarly, line 375.
Figure 4 why are the diet treatments not in the correct order from left to right?
Line 321. What does "concretely" mean?
Figure 5. Whay are there 4 datapoints for the G4 treatment when the others only have 3 and only 3 fish were sampled?
Line 357. The word "biomarker" is used in regards to microbiota but I do not think this is appropriate.
Line 368-371 is Discussion, not Results.
Discussion. I find the Discussion compares findings from too many dissimilar species to be useful. There is literature on termites, pigs, humans and human cell lines that has little relevance to fish.
Line 387 suggests that favism can be induced but it is caused by a genetic mutation and you either have it or you don't. It is not induceable.
Line 396 suggests "high broad bean content (about 50%) above the level in this study.." but this is unclear. 50% broad beans was used in your study, or do you mean 75% broad beans? Also, the difference in outcome from your study and Gaber et al 2006 may be because of the age/size of the fish or duration of the study. Small fish (as used by Gaber et al) are very different to 600g growout fish. Also, is it Gaber et al or Gaber 2006?
Line 417 you did not measure the T-AOC levels of the diets.
Also, this section makes me think about what is the appropriate group to call the 'control' in your study. Is it G1 or G6? If it's G6 (because I would assume that the fishfarm you sourced the fish from were using a commercial diet) then are the G6 data affected by the fact that one or more of the barrels of fish in that group had high mortality? If so, were the surviving fish stressed or diseased?
The Conclusions need to indicate the limitations of your study - that the diets were only fed for 90 days, and these were >600g fish. Line 563 I don't believe you looked at immune system parameters.
Comments on the Quality of English Language
Some spelling mistakes eg Table 1 "Permix" instead of "Premix"
Author Response
Comment: The wording of the title needs to be improved. This is not "effects of the growth performance..." but "effects on the growth performance...". Also, in every mention of the scientific name of Nile tilapia (apart from in the references) the spelling is incorrect - it is Oreochromis not Orechromis.
Reply: The title has been revised into ‘Effects of Broad Bean Diet on the Growth Performance, Muscle Characteristics, Antioxidant Capacity, and Intestinal Health on Nile Tilapia (Oreochromis niloticus)’. All spelling mistakes of the scientific name of Nile tilapia have been modified.
Comment: In this manuscript the word "crisp" appears to have several meanings. It refers to a diet (line 12), a dietary supplement (line 19), a marketing description of a fish with particular flesh texture (line 24) and fish flesh properties (line 43). This is confusing and the authors need to use different words for the diet. It is also unclear whether crispy fish is eaten raw or if it needs to be cooked. This needs to be addressed.
Reply: We are so sorry about the confusion caused by the word ‘crisp’. We have changed the wording again. We uniformly used the words ‘crispy’ and ‘crisping’, ‘crispy’ is an adjective to describe the fish flesh properties as ‘crisping’ is a present participle to describe the action of make the fish meat crispy.
Comment: There is no simple summary.
Reply: simple summary has been added.
Comment: The Abstract needs to say what size of tilapia were used and how long the fish were fed on the diets. Also, it needs to be stated whether the final weight of the tilapia is the typical size at which this species is harvested and marketed in the region. If it is not then then how appropriate is the study?
Reply: The information involved in the size of tilapia and feeding time has been added in the Abstract [Line 20]. Besides, the final weight of tilapia is the typical size at which this species is harvested and marketed in our region.
Comment: Lines 77 & 80 suggests the manuscript will deal with "breeding technique" and "breeding industry" but I did not read anything about this.
Reply: We are so sorry about the confusing words. We want to use ‘breeding’ to describe ‘culture’, it has been changed into ‘culture’ to avoid confusion.
Comment There is information missing about the manufacture of diets G1-G5. Line 96 talks about water being added and then a puffed diet is mentioned - but no indication of whether a cold compressed pellet or heated extruder was used.
Reply: We are so sorry about the lack of a related description. The puffed diets of tilapia used a heated extruder. The related description has been added in that line.
Comment: Were the fish mixed sex or all male/all female?
Reply: The fish mixed sex.
Comment: The diets are described but there is nothing about the nature of diet used by the farm to grow the fish to 600g before the trial started. The fish were then fed diet G1 for 7 days and then most were changed to diets G2-G6 for the remainder of the trial. The fish were not sampled (other than weight/length) at the start of the trial so any and all suggestions that the values of various parameters increased or decreased in the various groups over the course of the trial are not correct.
Reply: The fish fed by the unified commercial diet on the farm till the weight up to 600g, so they are in similar states. The fish feeding diet G1 for 7 days before the trial is a common domestication operation.
Comment: How were the fish fed - to satiation? was uneaten food collected and weighed? There's no suggestion that feed allocation was measured until we get to FCR. How was water quality maintained - flow through? Recirculation? Aeration? What was the cause of mortality?
Reply: ‘The fish fed-to satiation’ was adjusted by observing the number of residuals on the second day to adjust the amount of feeding on the day. The residuals will be collected and weighed every day. The fish were cultured in a barrel with aeration, and half of the sea water was replaced with fresh sea water every day. There was no significant difference among all groups in survival rate, the mortality was natural death.
Comment: Line 137. Where is "the fixed position" that the muscle tissue was removed from? There is no information about when or how the muscle tissue was stored/tested for texture profiling. Was it done immediately, or after rigor mortis, at 4 degrees or room temperature, raw, or after cooking?
Reply: “The fixed position” is the position directly below the foremost part of the dorsal fin of the back. The texture profiling was performed immediately after the muscle was dissected. The performing process was at room temperature using raw muscle material.
Comment: Line 181. There needs to be a reference for the "specific primers with a barcode" used for the microbiome analysis.
Reply: The information on ‘specific primers with a barcode’ has been added as a supplemental file [Table SI].
Comment: Lines 195 and 199 are contradictory. Were data analyzed by R or SPSS?
Reply: We are so sorry about the lack of the contradictory description here. The mistake has been revised.
Comment: Figure 1. There is nothing on the photographs to show which image refers to which diet treatment. The font used for the scale bar is too small. It is clear that the muscle fibers are not circular in cross section, so how can they have a diameter? How was the 'diameter' of the 3, 4 or 5-sided cross sectional area calculated? In Figure 2 are the graphs (A) and (B) the correct way around? The scale bar on the photographs would suggest the muscle fibres are about 200um 'square' not 10um as the graphs and text (line 256) would suggest. Also, if you estimate the cross sectional area of a 10um x 10um fiber = 100um2 and then multiply this by 130 (the suggested number of fibers per mm2) you get 13,000um2. This is only 1.3% of the 1,000,000um2 that is 1 mm2. If you swap the graphs, then the muscle fiber is 200 x 200 = 40,000um2 and if there are 10 per mm2 then it is 400,000um2 which is 40% of the area. If I am correct then I have serious concerns about the credibility of this aspect of the work. Also, I cannot see how G4 muscle fiber density can be shown as 'b' when it clearly overlaps so much with G1, G3 and G4. Please re-check.
Reply:
- The figure has been redrawn and presented.
- Muscle fiber diameter is calculated according to the common method in animals’ muscle research. The method assumes that the muscle fibers are cylindrical, the diameter was calculated according to the equation s = πr2 (where s and r are the muscle fiber area and radius, respectively).
Reference:
Ma, L.L., Kaneko, G., Wang, X.J., **e, J., Tian, J.J., Zhang, K., Wang, G.J., Yu, D.G., Li, Z.F., Gong, W.B. and Yu, E.M., 2020. Effects of four faba bean extracts on growth parameters, textural quality, oxidative responses, and gut characteristics in grass carp. Aquaculture, 516, p.734620.
- We are so sorry about the serious mistake in the muscle fiber diameter data. That was caused by the error in unit conversion (mm to um), the unit output from the software was mm, we converted it into um but made a mistake in the decimal point. All the mistakes in the data of the diameter and density of muscle fibers have been corrected.
Comment: Figure 3 is not easy to look at. Please consider putting the six 100x images together and the six 400x images together rather than mixing them. The font for the scale bar is too small. Table 5 - why are all the values in mm when the um will give you more decimal places?
Reply: Thank you for your advice. Figure 3 has been re-arranged and presented. The reason why use mm due to balance the decimal place of ‘Number of goblet cells per unit length’ with the decimal place of ‘villus epithelial length’, ‘lamina propria thickness’, ‘villous epithelial length’, and ‘muscular thickness’.
Comment: Lines 309 and 312 use "increased most" and "grew up(?)" when there was no time course for changes in microbiome. Line 340 suggests "was reduced after the fish were fed with G4-G6 for 90 days" but this is misleading since there was no measurement made at day 0. Similarly, line 375.
Reply: We are so sorry about the wrong presentation here. They have all been revised.
Comment: Figure 4 why are the diet treatments not in the correct order from left to right?
Reply: We are so sorry about the unclear and direct presentation here. We wanted to show the difference among treatment groups, so they were put in front.
Comment: Line 321. What does "concretely" mean?
Reply: It means ‘in detail’ here, we wanted to further show which groups were clustered together.
Comment: Figure 5. Whay are there 4 datapoints for the G4 treatment when the others only have 3 and only 3 fish were sampled?
Reply: We originally all used 3 fish samples per treatment to perform analysis, but you can find that the data points for the G4 treatment are more dispersed than other groups, especially the data point in the bottom right corner of the figure. Hence, one more sample was added to the analysis.
Comment: Line 357. The word "biomarker" is used in regards to microbiota but I do not think this is appropriate.
Reply: Thank you for your advice. Word ‘biomarker’ has been replaced by ‘Microorganisms’.
Comment: Line 368-371 is Discussion, not Results.
Reply: Thank you for your advice. The content has been adjusted.
Comment: Line 387 suggests that favism can be induced but it is caused by a genetic mutation and you either have it or you don't. It is not induceable.
Reply: The statement of ‘favism-inducing factors’ was proposed by many researches (Burbano et al., 1995; Jamalian, 1999; Lattanzio et al., 1982). It is indeed a genetic disease but it can indeed be induced by some substances. According to previous studies, Favism is a hemolytic anemia caused by the compounds Vicine and Convicine in G6PD insufficiency patients. It is a disease caused by hereditary abnormality, which is caused after the ingestion of foodstuffs containing faba beans. It is caused due to the high concentration of amino pyrimidine derivatives; divicine and convicine present in faba beans. Vicine and Convicine are the glycosides present in faba beans which are responsible for favism. Vicine and convicine can cause favism in individuals with a genetically inherited deficiency in G6PD by injecting.
References:
Burbano, C., Cuadrado, C., Muzquiz, M. and Cubero, J.I., 1995. Variation of favism-inducing factors (vicine, convicine and L-DOPA) during pod development in Vicia faba L. Plant foods for human nutrition, 47, pp.265-274.
Jamalian, J., 1999. Removal of favism‐inducing factors vicine and convicine and the associated effects on the protein content and digestibility of fababeans (Vicia faba L). Journal of the Science of Food and Agriculture, 79(13), pp.1909-1914.
Lattanzio, V., Bianco, V.V. and Lafiandra, D., 1982. High-performance reversed-phase liquid chromatography (HPLC) of favism-inducing factors in Vicia faba L. Experientia, 38(7), pp.789-790.
Comment: Line 396 suggests "high broad bean content (about 50%) above the level in this study.." but this is unclear. 50% broad beans was used in your study, or do you mean 75% broad beans? Also, the difference in outcome from your study and Gaber et al 2006 may be because of the age/size of the fish or duration of the study. Small fish (as used by Gaber et al) are very different to 600g growout fish. Also, is it Gaber et al or Gaber 2006?
Reply: We used 40% and 50% broad bean content as 25%, 50%, 75%, and 100% were used in Gaber’s research. We indicated one possibility that broad beans did not replace the added amount of fishmeal but rather replaced a portion of the plant-sourced protein in the study of Gaber. We agree that it’s possible that the difference could cause by the age/size of the fish or the duration of the study, related description has been added [Line 400-404]. In addition, the literature format error has been corrected (changed ‘Gaber et al’ into ‘Gaber 2006’).
Comment: Line 417 you did not measure the T-AOC levels of the diets.
Reply: The data of T-AOC levels was shown in Table 3. Or you think ‘the T-AOC levels of all broad bean diets were significantly lower than that of the control, meaning that the supplementation of broad beans weakened the antioxidant capacity of fish’ made you confuse? We changed the description ‘the T-AOC levels of all broad bean diets’ into ‘the T-AOC levels of all broad bean diets groups’ to avoid that.
Comment: Also, this section makes me think about what is the appropriate group to call the 'control' in your study. Is it G1 or G6? If it's G6 (because I would assume that the fishfarm you sourced the fish from were using a commercial diet) then are the G6 data affected by the fact that one or more of the barrels of fish in that group had high mortality? If so, were the surviving fish stressed or diseased?
Reply: The control was G1. G6 was used to compare the crisping effects with our designed diets. Indeed, the mortality of G6 was a little high, but there was no significant differences with other groups.
Comment: The Conclusions need to indicate the limitations of your study - that the diets were only fed for 90 days, and these were >600g fish. Line 563 I don't believe you looked at immune system parameters.
Reply: The limitations of this study has been added in the Conclusions. The immune system-related parameters in this study are ADA and LZM. There were no significant differences on LZM, The original description specifically wanted to illustrate no variation in LZM. To avoid confusion, the description has changed into ‘had no negative influence on liver health and the immune system’ to contain all immune system-related parameters.
Comment: Some spelling mistakes eg Table 1 "Permix" instead of "Premix"
Reply: Thank you for your reminding. The mistake here has been revised.